# Ganciclovir Resistance-Linked Mutations in the HCMV *UL97* Gene: Sanger Sequencing Analysis in Samples from Transplant Recipients at a Tertiary Hospital in Southern Brazil

**DOI:** 10.3390/diagnostics15020214

**Published:** 2025-01-18

**Authors:** Anna Caroline Avila da Rocha, Grazielle Motta Rodrigues, Alessandra Helena da Silva Hellwig, Dariane Castro Pereira, Fabiana Caroline Zempulski Volpato, Afonso Luís Barth, Fernanda de-Paris

**Affiliations:** 1Faculdade de Biomedicina, Universidade Federal de Ciências da Saúde de Porto Alegre, Porto Alegre 90050-170, Rio Grande do Sul, Brazil; 2LABRESIS–Laboratório de Pesquisa em Resistência Bacteriana, Hospital de Clínicas de Porto Alegre, Porto Alegre 90035-903, Rio Grande do Sul, Brazil; darianepereira@hcpa.edu.br (D.C.P.); albarth@hcpa.edu.br (A.L.B.); 3Programa de Vigilância em Saúde, Residência Integrada em Saúde da Escola de Saúde Pública do Rio Grande do Sul, Porto Alegre 90610-001, Rio Grande do Sul, Brazil; 4Programa de Pós-Graduação em Ciências Médicas, Universidade Federal do Rio Grande do Sul, Porto Alegre 90160-093, Rio Grande do Sul, Brazil; gmorodrigues@hcpa.edu.br (G.M.R.); ahellwig@hcpa.edu.br (A.H.d.S.H.); 5Serviço de Diagnóstico Laboratorial, Hospital de Clínicas de Porto Alegre, Porto Alegre 90035-903, Rio Grande do Sul, Brazil; 6Departamento de Biociências, Universidade Federal do Paraná, Setor Palotina, Palotina 85953-128, Paraná, Brazil; fabiana_volpato@yahoo.com.br

**Keywords:** cytomegalovirus, mutation, ganciclovir resistance, transplant recipients

## Abstract

**Background/Objectives:** Human cytomegalovirus (HCMV) DNAemia remains a significant concern for transplant recipients, largely due to mutations in the viral genome that may lead to antiviral-resistant strains. Mutations in the *UL97* gene are frequently associated with resistance to ganciclovir (GCV), highlighting the importance of early mutation detection to effectively manage viremia. This study aimed to optimize a Sanger sequencing protocol for analyzing GCV resistance-linked mutations in the HCMV *UL97* gene from plasma samples of transplant patients treated at Hospital de Clínicas de Porto Alegre, Rio Grande do Sul, Brazil. **Methods:** A nested-PCR approach combined with a touchdown PCR method was employed to enhance the sensitivity and specificity of the sequencing analysis. **Results:** The study sample included various transplants, encompassing solid organ and bone marrow recipients. Among 16 sequenced samples, 8 exhibited nucleotide substitutions resulting in amino acid changes. Notably, the A594V and C603W mutations, associated with GCV resistance, were identified in four samples. Additionally, three mutations with unknown phenotypic impact (P509L, A628T, and H662Y) and two viral polymorphisms (N510S and D605E) were detected. Furthermore, double peaks in the Sanger electropherograms, indicative of mixed viral populations of HCMV were observed in seven samples. **Conclusions:** The optimized Sanger sequencing protocol provides a cost-effective solution for detecting GCV resistance mutations in HCMV *UL97* among transplant recipients. This approach could improve the understanding of HCMV strain dynamics and serve as a valuable tool for long-term patient monitoring, particularly within resource-constrained settings such as the public health systems of middle-income countries.

## 1. Introduction

Human cytomegalovirus (HCMV or HHV-5—Human betaherpesvirus 5) DNAemia poses a substantial clinical challenge for transplant recipients, requiring vigilant monitoring strategies to mitigate post-transplant risks and complications [1,2]. This occurs due to the high prevalence and seropositivity of this virus within the global population, which poses significant risks of severe clinical manifestations for immunosuppressed individuals, such as transplant recipients [3]. To illustrate this widespread prevalence, studies conducted in the Brazilian population have reported approximately 96% seropositivity among blood donors [4,5].

Especially concerning is the emergence of HCMV strains with mutations, often located in the *UL97* gene, leading to resistance to first-line therapy such as ganciclovir (GCV). Alternative therapies, such as maribavir (MBV), have been investigated as promising options for managing refractory HCMV infections, with or without resistance to standard therapy, in post-transplant patients [6,7]. Despite these advancements, GCV continues as the standard treatment for viremia and currently remains the only antiviral available in the public health system in Brazil, highlighting a critical gap in therapeutic availability [2,8,9].

About 90% of the described cases of GCV resistance are linked to mutations in the *UL97* gene. Most mutations that impact pUL97 protein activity occur at codons 460 and 520, with mutations or deletions also occurring near codons 590–607. These alterations impair GCV phosphorylation, which is necessary for its antiviral activity, leading to an inadequate or failed therapeutic response [10,11,12,13].

In clinical suspicion surrounding the presence of GCV-resistant strains, applying laboratory techniques for mutation analysis becomes imperative, playing a crucial role in forecasting viremia outcomes and guiding treatment decisions. The absence of an early resistance diagnosis increases the healthcare burdens associated with patient management, especially among immunocompromised individuals such as transplant recipients [3,14]. HCMV remains the most significant infectious complication following solid organ transplantation, directly impacting both patient and graft survival through direct viral effects and indirect complications, such as superinfections and graft rejection [15].

Genotypic assays involve the direct analysis of HCMV genetic material, enabling the detection of specific mutations within the viral genome. However, protocols for screening viral mutations are not universally applied, particularly due to variations in laboratory infrastructure, funding for laboratory tests, and the training of technical staff. Sanger sequencing remains the standard method for amplifying target antiviral sequences, facilitating the identification of nucleotide changes linked to resistance [10,11]. Therefore, despite the availability of more advanced techniques such as next-generation sequencing (NGS), Sanger sequencing remains relevant, particularly in resource-limited settings like the public health systems of middle-income countries.

Despite the significance of data on HCMV resistance to antivirals, particularly given the virus’s crucial role in post-transplant outcomes [15], a knowledge gap persists in many regions worldwide and is especially evident in middle-income countries [8,16,17]. In this context, the present study aims to optimize the Sanger sequencing technique to identify GCV resistance-associated mutations in the HCMV *UL97* gene from plasma samples of transplant patients. The goal is to advance scientific knowledge on this subject and provide a potential tool for patient monitoring.

## 2. Materials and Methods

### 2.1. Clinical Samples and Ethical Considerations

The specimens were collected from HCMV-positive individuals undergoing solid organ and bone marrow transplants at a tertiary hospital in Porto Alegre, located in the southern region of Brazil. Plasma samples were obtained based on the viral load detected through qPCR using the Alinity™ m CMV assay (Abbott, Abbott Park, IL, USA) and were collected between July 2022 and June 2024. HCMV viral load results are reported in IU (international units), as per World Health Organization (WHO) guidelines [18]. In this work, we converted the values to copies/mL and reported both units.

A total of sixteen samples were sequenced, as outlined in Table 1. Clinical suspicion of GCV resistance typically arises when the HCMV viral load persists or increases, even after prolonged treatment. In cases where multiple samples were available from a single patient, we selected the one with the highest detected viral load. This approach maximized the likelihood of detecting resistance mutations, as higher viral loads may increase the probability of identifying genetic variants related to therapeutic failure [1,2,10]. This study was approved by the Ethics Committee of the Hospital de Clínicas de Porto Alegre under CAAE number 66786323300005327 and involved the evaluation of plasma samples.

### 2.2. DNA Extraction and Polymerase Chain Reaction (PCR) Amplification

The samples were stored at −20 °C until further processing [19] and viral DNA was extracted using two different methods: an automatized one, which used the *m2000*^®^ RealTime System (Abbott, Abbott Park, IL, USA), or a manual one, which used the QIAamp^®^ Viral RNA Mini Kit (Qiagen, Valencia, CA, USA), following the manufacturer’s instructions.

To amplify the regions of interest in the *UL97* gene, we employed a nested-PCR approach, using two specific primer sets outlined by Tasoujlu and colleagues [8]. For assay standardization, the AmpliRun^®^ Cytomegalovirus DNA Control was utilized (strain AD169) (Vircell, Santa Fe, Granada, Spain).

Briefly, the first PCR round was performed using the primers 1278F and 2013R, resulting in a 736-base pair (bp) PCR product. The reaction mix consisted of 20 μL of Platinum™ PCR SuperMix High Fidelity (Invitrogen, Waltham, MA, USA), 0.25 μL of each primer at 10 pM (1278F and 2013R), and 5 μL of extracted DNA. The cycling conditions for the first PCR round were as follows: 95 °C for 5 min, followed by 40 cycles at 94 °C for 1 min, annealing at 61 °C for 1 min and 30 s, and extension at 72 °C for 1 min, with a final extension step at 72 °C for 5 min.

For the second round, a different set of internal primers, 1292F and 1998R, was specifically designed to bind to the amplified product from the first round, resulting in a predicted fragment of 707 bp. A 2 μL aliquot of the first-round product was added to 23 μL of Platinum™ PCR SuperMix High Fidelity (Invitrogen, Waltham, MA, USA) and 0.25 μL of each primer at 10 pM (1292F and 1998R). The primer set used is shown in Table 2. The touchdown PCR method was applied to minimize off-target priming and increase specificity. The cycling conditions for the second round were as follows: 95 °C for 5 min, followed by 15 cycles of 94 °C for 1 min, annealing at 60 °C for 1 min and 30 s, and extension at 72 °C for 1 min; 10 cycles of 94 °C for 1 min, annealing at 58 °C for 1 min and 30 s, and extension at 72 °C for 1 min; and 10 cycles of 94 °C for 1 min, annealing at 56 °C for 1 min and 30 s, and extension at 72 °C for 1 min. A final extension step was performed at 72 °C for 5 min. Electrophoresis in agarose gel was conducted to validate the amplification of *UL97* fragments.

### 2.3. In Silico Analysis

To assess the universality of the employed primer sets for amplification and Sanger sequencing, and their ability to flank distinct clinical HCMV strains representative of natural infections [20], we adopted an in silico approach. HCMV *UL97* gene sequences from the Toledo (GU937742.2), TB40/E (KF297339.1), VR1814 (GU179289.1), AD169 (FJ527563.1), and Towne (FJ616285.1) strains were downloaded from GenBank. These sequences were aligned using the Unipro UGENE software v47.0 against the reference human cytomegalovirus *UL97* gene from the Merlin strain (NC_006273.2) and the AmpliRun^®^ Cytomegalovirus DNA Control (strain AD169) (Vircell, Santa Fe, Granada, Spain), representing the 707 bp fragment amplified by PCR.

### 2.4. Limit of Detection (LoD) and Repeatability

To determine the limit of detection, we evaluated a diluted clinical sample to achieve an HCMV viral load of 2800 copies/mL (4368 IU/mL or 3.6 log IU/mL). This value was chosen based on the lowest values observed among patients who obtained conclusive sequencing results. The final dilution was determined as the LoD was subjected to a series of 20 parallel PCRs to establish repeatability.

### 2.5. Bi-Directional Sanger Sequencing

The amplicon purification was carried out using the ExoSAP-IT PCR Product Cleanup enzyme (Affymetrix, Santa Clara, CA, USA) to remove residual unincorporated primers and nucleotides. Purified products were submitted to bi-directional Sanger sequencing using the BigDye™ Terminator v3.1 Cycle Sequencing Kit (Applied Biosystems, Foster City, CA, USA). Sequencing was processed on an ABI 3500 Genetic Analyzer (Applied Biosystems, Foster City, CA, USA).

The quality of the sequencing data was assessed through Sanger electropherograms for both forward and reverse sequences and analyzed with the CodonCode Aligner v.11.0.2 software. Each sequencing was evaluated using Phred quality scores. To enhance the accuracy of the analysis, two directions (forward and reverse) for each sample were aligned against the reference genome of the Merlin strain (GenBank: NC_006273.2). Using both sequences from each sample improves the reliability of the detected mutations, especially in cases where co-infection with mixed viral populations is suspected [21,22].

### 2.6. Data and Mutation Analysis

Following sequencing, data analysis was performed using the Unipro UGENE software v.47.0, with sequence alignment against the reference human cytomegalovirus genome (Merlin strain, GenBank: NC_006273.2) to identify nucleotide variants. The default software parameters were applied, including a trimming quality threshold of 30 and a minimum mapping similarity of 80%.

Two operators independently reviewed the identified nucleotide variants to ensure the robustness of the mutation analysis. The cross-verification of the results reduced the risk of subjective bias and increased the accuracy of the mutation identification. Variants were further analyzed for their potential link to GCV resistance, drawing on previously documented mutations associated with antiviral resistance and natural polymorphisms, using, as a reference, the Comprehensive Herpesviruses Antiviral Drug Resistance Mutation Database (CHARMD) developed by Tilloy and colleagues [23].

Furthermore, we used an algorithm developed by the AG Bioinformatics and Systems Biology Institute of Neural Information Processing/Institute of Virology (ULM University) to analyze the protein consequences of the identified mutations [24]. Co-infections involving HCMV strains, characterized by double viral populations, could be detected. This analysis was performed by examining Sanger sequencing electropherograms, following the methodology outlined by Castor and colleagues [25].

Registration for access to genetic heritage will be conducted through SISGEN (Sistema Nacional de Gestão do Patrimônio Genético e do Conhecimento Tradicional Associado), following the guidelines set by the Brazilian Genetic Heritage Management Council.

## 3. Results

The study sample included various types of transplants, encompassing both solid organ and bone marrow recipients. The specimens were collected from a tertiary hospital in southern Brazil and comprised kidney transplant recipients (43.75%), along with bone marrow (31.25%), liver (18.75%), and combined heart and kidney transplant recipients (6.25%).

The primer sets employed in the PCR and Sanger sequencing flanked key regions of the *UL97* gene across all HCMV strains analyzed in silico. Alignment of the primer sets with sequences from the Merlin (NC_006273.2), Toledo (GU937742.2), TB40/E (KF297339.1), VR1814 (GU179289.1), AD169 (FJ527563.1), and Towne (FJ616285.1) strains revealed the suitability of the primers binding in conserved regions of the pUL97 kinase domain. Most mutations reported in the literature as GCV-resistant commonly appear at codons 460 and 520, and codons 590–607 [26]. The 707 bp fragment, defined in this study, covered, on average, codons 445–670 of pUL97 (Figure 1).

The detection limit for the PCR protocol employed in this study was approximately 2800 copies/mL, equivalent to 4368 IU/mL or 3.6 log IU/mL. At this threshold, a repeatability assay with 20 parallel PCRs was performed, achieving a 95% amplification rate (19/20). These results indicate that in 95% of cases where patients exhibit HCMV plasma viral loads of 2800 copies/mL, Sanger sequencing can be successfully performed, yielding conclusive results.

Sequencing analysis yielded reliable data for the majority of samples. The Sanger sequencing protocol produced reliable results, with average read lengths of 640 bp for forward sequences and 662 bp for reverse sequences. The average Phred quality score across all samples was 66 for consensus sequences, indicating high-confidence calls with a low probability of errors—approximately one in 1,000,000—corresponding to a base call accuracy of 99.9999%. This underscores the quality and fidelity of the bidirectional sequencing process.

Despite the quality of sequencing data, there were two isolated cases where one of the sequence reads (either forward or reverse) exhibited slightly lower quality. Nevertheless, the single-direction in these cases still maintained Phred quality scores above 50, ensuring sufficient quality for the detection of relevant mutations in the *UL97* gene. These results confirmed the accuracy and robustness of the sequencing, even in cases where only one sequence direction was available.

Among the nucleotide substitutions in our samples, we identified substitutions in twelve of the sixteen samples, representing 75% of the total analyzed. Of these, approximately 33% were classified as synonymous mutations that did not result in amino acid changes in the pUL97. Missense mutations in the *UL97* gene fragment were identified in eight samples (50%), including P509L, N510S, A594V, D605E, C603W, A628T, and H662Y (Table 3). These mutations are located within the kinase domain of pUL97. Among them, A594V and C603W are canonical missense mutations previously associated with GCV resistance. In contrast, P509L, A628T, and H662Y have an unknown phenotype, with no reports in the literature regarding their impact on antiviral resistance to date. Additionally, N510S and D605E are GCV-sensitive viral polymorphisms.

Furthermore, in 43.75% (7/16) of the samples, the presence of double peaks in the Sanger electropherogram was observed (Figure 2). These instances were attributed to mixed viral variants with mutant and wild-type strains, which generated overlapping signals that complicated the base calling. In such cases, manual curation of the data was performed to differentiate the co-infecting viral variants and ensure accurate mutation identification. During this process, a detailed visualization of the electropherogram was conducted to identify overlapping peaks. Peaks were deemed acceptable for “true” base calling if their height was less than 20% of the main sequence peak. Otherwise, such peaks were disregarded due to the low quality and unreliability caused by the interference [22]. This additional validation step ensured that no significant variants were overlooked due to potential sequencing errors, further reinforcing the reliability of the data.

## 4. Discussion

This study aimed to evaluate the performance of Sanger sequencing and its effectiveness in detecting mutations in the *UL97* gene. The Sanger method may be of particular relevance in resource-limited settings, such as public health systems in middle-income countries. Sanger sequencing is widely used for genotyping and identifying HCMV GCV-resistant mutations. Studies in other clinical contexts reinforce its applicability and reliability for routine diagnostics [3].

Most of the specimens analyzed in this study were obtained from kidney transplant recipients, comprising 43.75% (7/16). However, HCMV DNAemia is not exclusive to kidney transplant recipients, as our sample also included other transplant patients with HCMV detected in peripheral blood (see Table 1). These findings underscore the importance of ensuring access to mutation testing for all transplant recipients [3,14,27].

In silico analysis showed the primer set binding conservation sites across all analyzed strains. These findings suggest that the employed primers are suitable for detecting clinically significant mutations in a wide range of HCMV strains. Testing multiple strains is relevant due to the genomic diversity observed among HCMV isolates [28,29]. The amplified region covered mutations previously associated with GCV resistance within the ATP-binding domain of the pUL97 kinase.

Beyond in silico analysis, establishing the limit of detection (LoD) using viral load values is crucial for validating the test’s clinical application in the care of transplant patients [30]. The LoD for the PCR protocol established in this study was 2800 copies/mL (equivalent to 4368 IU/mL or 3.6 log IU/mL). This threshold was sufficient to meet the needs of the patients included in this research and reliably yield conclusive results through Sanger sequencing. LoD values can vary significantly between different tests. This value reported is comparable to that observed by Mallory and colleagues [31] for their Sanger sequencing protocol used to detect resistance-associated mutations in HCMV. On the other hand, according to Kotton and Kamar (2023) [2], clinical signs of severe disease are more indicative of a potential resistance to antivirals than levels of HCMV DNAemia. Therefore, the sequencing LOD serves to guide attending physicians by indicating the minimum viral load that transplant recipients must present for effective sequencing test results.

The protocol described in this study achieved an average read length of 640 bp for forward sequences and 662 bp for reverse sequences, providing adequate coverage of the pUL97 kinase domain. The Phred quality scores obtained for our sequences were sufficiently high to ensure reliable results. The average Phred quality score across all samples was 66 for consensus sequences, reflecting high-confidence base calls with a low error probability. The reliability of the reported results was further reinforced through independent analyses by two operators and the use of bi-directional sequencing (sense and antisense strands).

The frequent detection of GCV-resistant missense mutations, such as A594V and C603W, underscores the Sanger method’s effectiveness in identifying clinically significant variants [32]. These mutations are well-documented in cases of therapeutic failure, as they affect critical regions of pUL97, particularly the ATP-binding domain. However, different mutations at different codons of the pUL97 confer varying amounts of resistance to GCV [33,34,35]. Changes at codons 594 and 603 were associated with 3- to 9-fold increases in the 50% inhibitory concentrations of GCV [33]. Their frequent occurrence highlights the need for continuous surveillance and early detection in transplant patients to guide therapeutic decisions and optimize clinical management [3,36,37].

Additionally, P509L, A628T, and H662Y mutations were identified in clinical samples. These viral genetic alterations have not been reported in the literature in association with GCV resistance and are classified as “non-database mutations” by the algorithm developed by ULM University [24]. Although not conclusively linked to resistance, these mutations may represent novel polymorphisms with potential clinical relevance, particularly in settings where alternative antiviral therapies are limited. Conversely, we detected N510S and D605E, GCV-sensitive viral polymorphisms, and synonymous mutations that did not result in amino acid changes but contributed to the genetic diversity of HCMV strains [37,38,39]. These alterations reflect viral evolution and underscore the genetic variability in HCMV.

Furthermore, samples with mixed viral populations, indicated by double peaks in the Sanger electropherograms, were identified. These co-infections with mixed viral variants, including both mutant and wild-type strains, present a challenge for sequencing and interpretation. Nonetheless, their detection and monitoring are clinically meaningful, as the recombination of mixed viral variants may contribute to the emergence of resistant strains, further complicating treatment strategies [12,40]. The occurrence of multiple HCMV strains has been well-documented in immunocompromised hosts, particularly transplant recipients. This phenomenon can be attributed to several scenarios, including reinfection of HCMV-seropositive recipients with novel strains transmitted by the donors post-transplantation and susceptibility to multiple exposures over time [41,42].

The results obtained in this study demonstrate the ability of Sanger sequencing to detect mixed viral populations in such cases. The presence of double peaks in nearly half of the analyzed samples (7/16) underscores the importance of evaluating and interpreting this information accurately. Although Sanger sequencing has inherent limitations in resolution compared to next-generation sequencing (NGS), manual curation and careful interpretation were employed with these samples, ensuring the accurate identification of mutations and differentiation of viral variants. This reinforces the importance of experienced operators in effectively managing complex samples.

Resistance evaluation is essential in post-transplant patient care, as treatment failure could result from either host-related factors or antiviral resistance. The genotypic resistance test is pivotal in differentiating between these two scenarios [43]. 

In our tertiary hospital, GCV is the standard treatment for HCMV viremia. Foscarnet (FOS) is reserved as an alternative for cases of resistance due to its limited availability, high cost, and significant adverse effects. Genotypic testing may confirm GCV resistance, aiding in the procurement and approval of FOS. 

Recently, MBV has been proposed for incorporation into the Brazilian Unified Health System (SUS) to manage post-transplant cases that are refractory or resistant to GCV. This underscores the importance of routine HCMV resistance testing to optimize therapeutic decision-making and improve transplant patient care. Although the genotypic results were not directly validated through phenotypic tests, to determine the effect of a specific mutation on GCV susceptibility, the interpretation of the data is grounded in the available scientific literature and supported by the information provided in the database CHARMD, developed by Tilloy and colleagues [23]. This database served as a reference for classifying the mutations identified in this study and ensuring their contextualization within the broader HCMV antiviral resistance research framework.

Overall, this study demonstrated the utility of Sanger sequencing as a reliable method for the timely detection of GCV-resistant *UL97* mutations in HCMV strains from clinical samples of transplant recipients. We have evaluated 16 samples to date, but we expect to expand both the sample size and the number of viral genomic regions sequenced. This method offers advantages in cost-effectiveness and accessibility, making it suitable for implementation in public health systems, including those in middle-income countries, to establish an appropriate approach for routine clinical screening of transplant patients suspected of GCV-resistant *UL97*. Early detection of these mutations is crucial for optimizing treatment strategies in transplant patients, potentially improving clinical outcomes and reducing the financial burden on healthcare systems. Expanding the analysis to include other genomic regions and integrating NGS with Sanger sequencing could provide a more comprehensive view of viral genetic diversity and resistance mechanisms. Such efforts would contribute to improved patient management and the development of more effective therapeutic strategies for managing post-transplant HCMV infections.

## 5. Conclusions

Early detection and continuous surveillance of the *UL97* gene to monitor HCMV strains in transplant patients are paramount for effective management. This approach benefits patients and the healthcare system by facilitating timely adjustments to antiviral therapies and reducing the risk of treatment failure. Sequencing of the *UL97* gene by the Sanger method provides valuable insights into the genetic dynamics of HCMV strains and remains a promising tool for monitoring and managing infection in transplant patients.

## Figures and Tables

**Figure 1 diagnostics-15-00214-f001:**
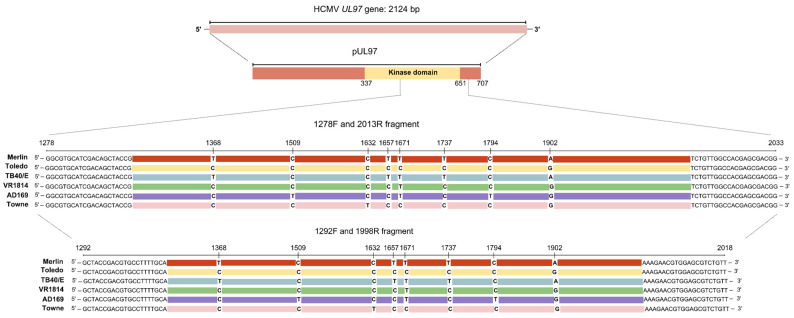
Schematic representation of the HCMV *UL97* gene and its pUL97 kinase domain located between codon positions 337 and 707. The primer sets used for amplifying these regions are labeled on the respective fragments. The 1278F/2013R and 1292F/1998R primer sets comprise all sequences from different HCMV strains (Merlin, Toledo, TB40/E, VR1814, AD169, and Towne). The alignment of sequences highlights strain-specific genetic variability.

**Figure 2 diagnostics-15-00214-f002:**
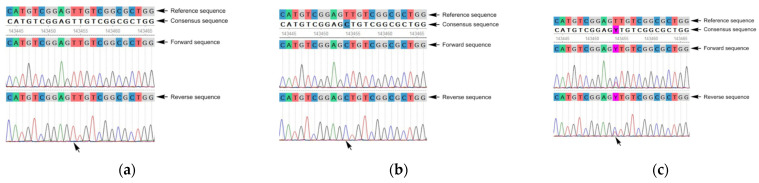
HCMV *UL97* gene mutation detection analysis by the Unipro UGENE software. The arrow indicates that it is located at the same genomic position. (**a**) Sequencing results of samples without mutations in the *UL97* gene. (**b**) Sequencing results of samples with synonymous mutations. (**c**) Sequencing results of mixed samples of two HCMV single nucleotide variants.

**Table 1 diagnostics-15-00214-t001:** Characteristics of samples collected and DNA extraction methods.

Sample ID	Transplant Type	Viral Load (Copies/mL; IU/mL)	Period of Collection	Method of DNA Extraction
A2	Kidney	3272; 5104	07/2022	m2000
A4	Kidney	8304; 12,954	07/2022	m2000
A5	Kidney	2964; 4624	07/2022	m2000
A6	Kidney	4994; 7791	07/2022	m2000
A10	Liver	46,544; 76,798	08/2022	m2000
A17	Liver	3424; 5341	09/2022	m2000
A20	Bone marrow	2962; 4621	09/2022	m2000
A21	Heart and kidney	108,903; 169,889	09/2022	m2000
A26	Bone marrow	6196; 9666	09/2022	m2000
A37	Kidney	6795; 10,600	11/2022	QIAmp
A39	Liver	13,510; 21,076	11/2022	QIAmp
A44	Kidney	156,554; 244,224	11/2022	QIAmp
A46	Kidney	121,222; 189,106	01/2023	QIAmp
A48	Bone marrow	86,809; 135,422	03/2023	QIAmp
A53	Bone marrow	65,958; 102,894	04/2024	QIAmp
A54	Bone marrow	111,697; 174,247	06/2024	QIAmp

**Table 2 diagnostics-15-00214-t002:** List of primers used in this study for PCR and Sanger sequencing.

PCR Round	Primer	Sequence (5′-3′)	Size (bp)
First reaction	1278F	GGCGTGCATCGACAGCTACCG	736
	2013R	CCGTCGCTCGTGGCCAACAGA	
Second reaction	1292F	GCTACCGACGTGCCTTTTGCA	707
	1998R	AACAGACGCTCCACGTTCTTT	

**Table 3 diagnostics-15-00214-t003:** The *UL97* mutations and polymorphisms.

Sample ID	Genomic Annotation	Amino Acid Change	GCV Resistance
A4	A143.326G	N510S	Genetic polymorphism ^1^
A5	C143.323T	P509L	Unknown mutation ^2^
A6	C143.578T	A594V	Mutation associated with drug resistance ^3^
	C143.781T	H662Y	Unknown mutation ^2^
A17	C143.612G	D605E	Genetic polymorphism ^1^
A44	C143.578T	A594V	Mutation associated with drug resistance ^3^
	C143.606G	C603W	Mutation associated with drug resistance ^3^
A48	C143.606G	C603W	Mutation associated with drug resistance ^3^
A53	C143.578T	A594V	Mutation associated with drug resistance ^3^
A54	G143.679A	A628T	Unknown mutation ^2^

GCV—Ganciclovir. ^1^ Reported as GCV-sensitive viral polymorphism. ^2^ Mutation not reported in the literature as associated with drug resistance. ^3^ Mutations confirmed to confer resistance by phenotypic testing and validated through gene marker transfer studies in vitro.

## Data Availability

The raw data of this study are available by request to the corresponding author, including the sequencing raw data.

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
