# Peer review of "Ganciclovir Resistance-Linked Mutations in the HCMV UL97 Gene: Sanger Sequencing Analysis in Samples from Transplant Recipients at a Tertiary Hospital in Southern Brazil"

_diagnostics, 2025, doi:10.3390/diagnostics15020214_

Round 1

Reviewer 1 Report

Comments and Suggestions for Authors

Management of HCMV infection is a critical issue for transplant recipients, and this study has the potential to contribute significantly to the field. The use of the nested PCR approach to enhance the sensitivity and specificity of sequencing analysis is impressive. Adding a few additional details is expected to further improve the quality of the paper.

Introduction

1. What are the CMV prevalence or seropositivity rates in Brazil?

2. Regarding the sentence, "A knowledge gap persists in many regions worldwide and is especially evident in middle-income countries"

-> it would be beneficial to include further explanation or references to support this statement.

Method

"To amplify the regions of interest in the UL97 gene, we employed a nested-PCR approach, using two specific primer sets outlined by Tasoujlu and colleagues [5]."

–> What are the differences between the referenced study and the methodology used in this paper?

Results

In Table 3, it would be helpful to include the treatment course of outcomes of patients

"such cases, manual curation of the data was performed to differentiate the co-infecting 234 viral variants and ensure accurate mutation identification [17]."

–> Could you provide more detailed explanation in this process?

Discussion

1. The advantages of this protocol compared to existing methods need to be emphasized more clearly.

2. Sixteen samples seem relatively few. Please discuss the potential for expanding the study with a larger sample size.

3. Regarding the LOD (limit of detection), is the value of 2,800 sufficient? A comparison with values reported in other studies would be helpful.

4. It would be useful to include an explanation in the Introduction or Discussion regarding why there is currently no universal protocol for ganciclovir mutation testing.

Author Response

Dear Reviewer,

Thank you for your valuable comments and suggestions. Below, we provide our responses to your questions and remarks. We sincerely believe that your feedback has significantly enhanced the scientific quality of the manuscript.

Sincerely,

Fernanda-de Paris

QUESTIONS AND ANSWERS:

Introduction

  1. What are the CMV prevalence or seropositivity rates in Brazil?

Answer = Thank you very much for raising this important point regarding HCMV seropositivity in the population. This aspect was indeed not addressed in the original version of the article. HCMV seropositivity in Brazil is notably high, with rates reaching up to 96% (Souza et al., 2010; Oliveira et al., 2022). 

To address this issue, we have added the following sentence to the Introduction in reviewed manuscript: “This occurs due to the high prevalence and seropositivity of this virus within the global population, which poses significant risks of severe clinical manifestations for immunosuppressed individuals, such as transplant recipients [3]. To illustrate this widespread prevalence, studies conducted in the Brazilian population have reported approximately 96% seropositivity among blood donors [4,5].” These sentences added in the reviewed manuscript were highlighted.

  1. Regarding the sentence, "A knowledge gap persists in many regions worldwide and is especially evident in middle-income countries"

-> it would be beneficial to include further explanation or references to support this statement.

Answer: Thank you for your observation. We have incorporated references to this sentence to provide additional support for our claim. The references have been highlighted in the revised manuscript.

Method

"To amplify the regions of interest in the UL97 gene, we employed a nested-PCR approach, using two specific primer sets outlined by Tasoujlu and colleagues [5]."

–> What are the differences between the referenced study and the methodology used in this paper?

Answer: Thank you for your question. In our study, we employed a Touchdown PCR approach for the second reaction of the nested PCR protocol to optimize the results with the available reaction mix. We utilized a high-fidelity DNA polymerase in our amplification protocol, ensuring accuracy and reliability. Unlike the purification method described by Tasoujlu et al., which utilized the AccuPrep PCR DNA Purification Kit, our protocol involved enzymatic purification. Additionally, our study encompasses a broader range of transplant patients treated at our institution, extending beyond kidney transplant recipients.

Results

In Table 3, it would be helpful to include the treatment course of outcomes of patients

Answer: Thank you for your note. Including the treatment course and patient outcomes in Table 3 would indeed provide valuable insights. However, this information was unavailable, as noted in the 'Informed Consent Statement' section.

"such cases, manual curation of the data was performed to differentiate the co-infecting 234 viral variants and ensure accurate mutation identification [17]."

–> Could you provide more detailed explanation in this process?

Answer: Certainly, we can provide additional details about the manual curation process. In general, sequence chromatograms can be refined through manual review, ensuring no overlapping peaks are present. Overlapping peaks may indicate the presence of closely related genotypes or non-specific primer binding to genomic material on the array (competing sequences, also referred to as 'background noise'). To distinguish between these two scenarios, a competing peak is considered acceptable if its height is less than 20% of the main sequence peak. If the concurrent peak exceeds this threshold, the entire sequence must be reviewed or discarded due to interference, as it may result from a low-quality and unreliable sequence.

To clarify this process in our manuscript, we included the following sentence: 'During this process, detailed visualization of the electropherogram was conducted to identify overlapping peaks. Peaks were deemed acceptable for "true" base calling if their height was less than 20% of the main sequence peak. Otherwise, such peaks are disregarded due to the low quality and unreliability caused by interference.' These revisions have been highlighted in the reviewed manuscript.

Discussion

  1. The advantages of this protocol compared to existing methods need to be emphasized more clearly.  

Answer: We agree with your point. To emphasize these advantages, we included the following sentence at the end of the discussion: “The method offers advantages in cost-effectiveness and accessibility, making it suitable for implementation in public health systems, including those in middle-income countries, to establish an appropriate approach for routine clinical screening of transplant patients suspected of GCV-resistant UL97.”.

  1. Sixteen samples seem relatively few. Please discuss the potential for expanding the study with a larger sample size.

Answer: Thank you for your comment. We plan to increase the sample size and investigate changes in other viral genes in a broader study. To clarify this for potential readers, we have added the following sentence: 'We have evaluated 16 samples to date, but we expect to expand both the sample size and the number of viral genomic regions sequenced.”

  1. Regarding the LOD (limit of detection), is the value of 2,800 sufficient? A comparison with values reported in other studies would be helpful.

Answer: Thank you for this note. The LOD is indeed crucial for the implementation of genotypic resistance testing for ganciclovir in transplant patients. However, there is no consensus on the specific viral load value associated with resistance to anti-HCMV drugs. According to Kotton & Kamar (2023) (bibliographic reference 2 in our manuscript), clinical signs of severe disease are more indicative of potential resistance to antivirals than CMV DNAemia levels. Therefore, the LOD serves to guide attending physicians by indicating the minimum viral load required for effective sequencing test results. Additionally, the study by Mallory et al. demonstrated an LOD of 3.2 log IU/mL for the Sanger-based HCMV genotypic resistance test they developed, while the value described in our study corresponds to 3.6 log IU/mL.

To clarify these points in the manuscript, we have added the following sentence in the LOD discussion: “On the other hand, according to Kotton & Kamar (2023) [2], clinical signs of severe disease are more indicative of potential resistance to antivirals than levels of CMV DNAemia. Therefore, the sequencing LOD serves to guide attending physicians by indicating the minimum viral load that transplant recipients must present for effective sequencing test results.”

  1. It would be useful to include an explanation in the Introduction or Discussion regarding why there is currently no universal protocol for ganciclovir mutation testing.

Answer: This is an important observation. Thank you for your comment. To clarify this point, we have added the following sentence to the Introduction: “However, protocols for screening viral mutations are not universally applied, particularly due to variations in laboratory infrastructure, funding for laboratory tests, and the training of technical staff.”

Reviewer 2 Report

Comments and Suggestions for Authors

HCMV is one of the most common and serious pathogens in organ transplant recipients in immunosuppressed states, and the widespread use of antiviral drugs has made HCMV resistance a serious clinical problem to be faced.

The most common pathway for HCMV resistance is through mutations in the HCMV UL97 gene. Sequencing is generally considered to be the more objective method, obtaining more comprehensive information.

The paper starts from infection samples of clinical real world transplant patient, sequencing and analyzing the HCMV UL97 gene mutation to detect mutations of drug resistance. It is conducive to the optimization of subsequent treatment regimens and the study of mechanisms of viral drug resistance.

The following questions are limited to academic discussion and do not affect the evaluation of the paper.

1.     the time span of the samples begins in 2022, were the storage conditions always -20°C (Line 98)? Did the freezing and thawing of the samples have an effect on the assay results?

2.     the lower limit of detection in the paper was determined based on sequencing results (Line 141). Are there any samples below the lower limit of detection where the CPE phenomenon was observed or viruses were detected after performing experiments of infection in cell model?

3.     Does the co-infection (Line 171) and double peaks (Line 231) mentioned in the paper imply the presence of mixed or non-dominant strains? How effective is the detection of these strains by the method described in the paper?

4.     Are the mutations mentioned in the paper (Line 224) resistant in cellular infection models? or what is the effect on the action of the drugs like ganciclovir?

Author Response

Dear Reviewer,

Thank you for your valuable comments and suggestions. Below, we provide our responses to your questions and remarks. We sincerely believe that your feedback has significantly enhanced the scientific quality of the manuscript.

Sincerely,

Fernanda-de Paris

QUESTIONS AND ANSWERS:

  1.     the time span of the samples begins in 2022, were the storage conditions always -20°C (Line 98)? Did the freezing and thawing of the samples have an effect on the assay results?

Answer: This is an important observation. We appreciate this question. The clinical specimens used in this study were residual material that the laboratory would have otherwise discarded after HCMV viral load testing. While it is known that freeze-thaw cycles can theoretically impact nucleic acid quality, previous studies, such as those by Jerome and colleagues, have demonstrated that both original specimens and extracted DNA stored for extended periods remain suitable for successful PCR amplification. This reference has been included in the paper bibliography (bibliographic reference 19 in our manuscript).

To ensure the integrity of the samples, we minimized the number of freeze-thaw cycles. DNA extraction was performed promptly after sample collection, and subsequent nested-PCR amplification and sequencing were conducted without evidence of compromised assay performance. The successful amplification and sequencing results obtained throughout this study suggest that the integrity of the extracted DNA was effectively preserved.

  1.     the lower limit of detection in the paper was determined based on sequencing results (Line 141). Are there any samples below the lower limit of detection where the CPE phenomenon was observed or viruses were detected after performing experiments of infection in cell model?

Answer: Thank you for raising this point. In this study, the lower limit of detection (LoD) was defined as 2,800 copies/mL (4,368 IU/mL or 3.6 log IU/mL) based on the ability to generate conclusive Sanger sequencing data. However, no experiments involving CPE were performed to evaluate samples with viral loads below this threshold. The focus of our research was restricted to genotypic analysis using clinical samples with detectable HCMV DNAemia above the LoD.

  1.     Does the co-infection (Line 171) and double peaks (Line 231) mentioned in the paper imply the presence of mixed or non-dominant strains? How effective is the detection of these strains by the method described in the paper?

Answer: Thank you for this pertinent question. Co-infections involving HCMV strains, as indicated by double peaks in Sanger sequencing electropherograms, are reflective of mixed viral populations. This phenomenon has been well-documented in immunocompromised patients, particularly transplant recipients. The findings in this study align with observations by Arav-Boger et al., Puchhammer-Stöckl et al., and Görzer et al., who described the occurrence of multiple HCMV genotypes in transplant patients and highlighted their potential for genetic recombination.  

In response to this question, we have clarified this topic in the revised manuscript and included references to Puchhammer-Stöckl et al. and Görzer et al. in the bibliography (bibliografic reference 41 and 42). The following additions were made to the Results: “Additionally, in 43,75% (7/16) of the samples, the presence of double peaks in Sanger electropherogram was observed (Figure 2). These instances were attributed to mixed viral variants with mutant and wild-type strains, which generated overlapping signals that complicated the base calling. In such cases, manual curation of the data was performed to differentiate the co-infecting viral variants and ensure accurate mutation identification. During this process, a detailed visualization of the electropherogram was conducted to identify overlapping peaks. Peaks were deemed acceptable for "true" base calling if their height was less than 20% of the main sequence peak. Otherwise, such peaks are disregarded due to the low quality and unreliability caused by interference [22].” 

And Discussion section: “These co-infections with mixed viral variants, including mutant and wild-type strains, present a challenge for sequencing and interpretation. Nonetheless, their detection and monitoring are clinically meaningful, as the recombination of mixed viral variants may contribute to the emergence of resistant strains, further complicating treatment strategies [12,40]. The occurrence of multiple HCMV strains has been well-documented in immunocompromised hosts, particularly transplant recipients. This phenomenon can be attributed to several scenarios, including reinfection of HCMV-seropositive recipients with novel strains transmitted by donor post-transplantation and susceptibility to multiple exposures over time [41,42]. The results obtained in this study demonstrate the ability of Sanger sequencing to detect mixed viral populations in such cases. The presence of double peaks in nearly half of the analyzed samples (7/16) underscores the importance of evaluating and interpreting this information accurately. Although Sanger sequencing has inherent limitations in resolution compared to next-generation sequencing (NGS), manual curation and careful interpretation were employed with these samples, ensuring the accurate identification of mutations and differentiation of viral variants.” These sentences added in the reviewed manuscript were highlighted.

  1.     Are the mutations mentioned in the paper (Line 224) resistant in cellular infection models? or what is the effect on the action of the drugs like ganciclovir?

Answer: Thank you for your question. Within the scope of this study, the identified mutations were interpreted based on the existing literature on genotypic and antiviral resistance. The mutations described in Line 224 (now updated to line 233 following revisions) are also detailed in the Discussion section. Previous phenotypic studies have demonstrated varying amounts of resistance to ganciclovir for specific mutations. While the genotypic results presented in this paper have not been directly validated using resistant cellular infection models, their interpretation is grounded in the available literature. This clarification has been incorporated in Materials and Methods into the revised paper: “Variants were further analyzed for their potential link to GCV resistance, drawing on previously documented mutations associated with antiviral resistance and natural polymorphisms, using as a reference the Comprehensive Herpesviruses Antiviral Drug Resistance Mutation Database (CHARMD), developed by Tilloy and colleagues [23].” And in Discussion, to further underscore this point: “Although the genotypic results were not directly validated through phenotypic tests, to determine the effect of specific mutation on GCV susceptibility, the interpretation of the data is grounded in the available scientific literature and supported by the information provided in the database CHARMD, developed by Tilloy and colleagues [23]. This database served as a reference for classifying the mutations identified in this study and ensuring their contextualization within the broader HCMV antiviral resistance research framework.” These sentences added in the reviewed manuscript were highlighted.

Round 2

Reviewer 1 Report

Comments and Suggestions for Authors

.